# The Enzymatic and Non-Enzymatic Function of Myeloperoxidase (MPO) in Inflammatory Communication

**DOI:** 10.3390/antiox10040562

**Published:** 2021-04-05

**Authors:** Yulia Kargapolova, Simon Geißen, Ruiyuan Zheng, Stephan Baldus, Holger Winkels, Matti Adam

**Affiliations:** Department III of Internal Medicine, Heart Center, Faculty of Medicine and University Hospital of Cologne, 50937 North Rhine-Westphalia, Germany; simon.geissen@uk-koeln.de (S.G.); zheng@smail.uni-koeln.de (R.Z.); stephan.baldus@uk-koeln.de (S.B.); matti.adam@uk-koeln.de (M.A.)

**Keywords:** myeloperoxidase, oxidative burst, NETs, cellular internalization, immune response, cancer, neurodegeneration

## Abstract

Myeloperoxidase is a signature enzyme of polymorphonuclear neutrophils in mice and humans. Being a component of circulating white blood cells, myeloperoxidase plays multiple roles in various organs and tissues and facilitates their crosstalk. Here, we describe the current knowledge on the tissue- and lineage-specific expression of myeloperoxidase, its well-studied enzymatic activity and incoherently understood non-enzymatic role in various cell types and tissues. Further, we elaborate on Myeloperoxidase (MPO) in the complex context of cardiovascular disease, innate and autoimmune response, development and progression of cancer and neurodegenerative diseases.

## 1. Introduction. MPO Conservation Across Species, Maturation in Myeloid Progenitors, and its Role in Immune Responses

Myeloperoxidase (MPO) is a lysosomal protein and part of the organism’s host-defense system. MPOs’ pivotal function is considered to be its enzymatic activity in response to invading pathogenic agents. During infection, the bactericidal and fungicidal properties of MPO contribute to pathogen clearing inside the phagosome.

Mature MPO is a basic metalloprotein with a molecular weight of 140 kilodaltons (kD), composed of two heavy and two light chains of approximately 55 and 13.5 kDa, respectively. MPO is a member of a diverse protein family, which is comprised of myeloperoxidase, eosinophil peroxidase, thyroid peroxidase, salivary peroxidase, lactoperoxidase, ovoperoxidase, peroxidasin, and others. The members of this family share 500 amino acid residues in length [1]. Interestingly, the first-ever orthologue of MPO can be found in the Australian saltwater crocodile (*Crocodylus porosus*), in which 39,19% of the orthologue sequence match the human MPO sequence [2]. Mouse and human MPO share 85% amino acid identity (90% homology) [3].

Upon protein synthesis, the MPO polypeptide matures through several steps (Figure 1). During the first step of maturation, the protein undergoes a series of proteolytic cleavages and posttranslational modifications, for example, N-glycosylation, and is converted from 80 kDa MPO into 90 kDa apoproMPO [4]. Two chaperones, calreticulin (CRT) and calnexin (CLN), are important for apoproMPO maturation. Subsequently, apoproMPO acquires a heme group while maturing in the endoplasmic reticulum to proMPO [5]. The inhibition of heme synthesis by succinyl acetone results in a maturation arrest of MPO [6]. Finally, in the last step of MPO maturation, the N-terminal protein domain is cleaved by proconvertase and intramolecular proteolytic cleavage leads to the formation of heavy and light chains of the enzyme. The mature protein is transported into the azurophilic granules where it is stored [7,8], whereas the remaining proMPO is secreted from the cell. MPO is the only member of the peroxidase-cyclooxygenase superfamily, which functions as a mature dimer, yet each hemi-MPO is enzymatically active. Neutrophils circulating in the blood have circadian oscillations of cell granularity. The maximal abundance of MPO-positive granules is observed in mice at midnight. At this time, neutrophils leave the bone marrow and decrease with the time in circulation. Plasma elastase activity, on the other hand, increases with time of circulation, furthermore suggesting the granule contents released over time [9].

MPO uses hydrogen peroxide and halide (Cl^−^, Br^−^, I^−^) or pseudohalide (SCN^−^) ions to catalyze the production of hypochlorous (HOCl), hypobromous (HOBr), hypoiodous (HOI) or hypothiocyanous (HOSCN) acids [11,12,13,14,15]. The enzyme is capable of catalyzing two types of redox reactions, specifically the halogenation cycle and the peroxidase cycle. The first step of the halogenation and peroxidase cycles are similar. During this step, the hydrogen peroxide reacts with native MPO, subsequently forming compound I, a ferryl form, which possesses a higher reduction potential compared to the ferric form of ground state MPO. During the second step of the halogenation cycle, the two electrons are transferred onto the halides, catalyzing acid formation. Subsequently, in the halogenation cycle, compound I is reduced to the native enzyme. As Cl^−^ is the most abundant halide in the blood, the halogenation reaction with Cl^−^ happens most frequently and hypochlorous acid (HOCl) was identified as the main product of MPO reacting with hydrogen peroxide. In the peroxidation cycle, the reduction of compound I is catalyzed by two successive one-electron transfer events and via a formation of compound II. In this case possible electron donors are aromatic acids [16], indole derivatives [17], nitrite [18], sulfhydryls [19], hydrogen peroxide [20] and other species.

The accumulated oxidative MPO products are highly reactive and released into the surrounding tissue (reviewed in [21,22]). This causes further oxidation of proteins and small molecules. Cysteine and methionine residues of proteins are most reactive, their oxidation leads to the formation of 3-Chlorotyrosine and Methionine sulfoxide at high concentrations of HOCl. Chloramines are weaker, however more stable, oxidants and can decompose to form aldehydes and low molecular-weight chloramines [23]. Reactive intermediates such as tyrosyl radical and nitrogen dioxide (NO_2_) are capable of initiating oxidation of lipids in plasma. Lipid and lipo-protein peroxidation at sites of inflammation by reactive intermediates of MPO oxidation is well known [24,25,26]. For example, the products of lipid peroxidation of arachidonic acid and linoleic acid were observed in inflammatory fluids from wild-type mice, but not in MPO-deficient animals [24].

MPO enzymatically promotes the formation of superoxides and thereby increases microbicidal potency primarily of the innate immune response. However, the contribution and essential role of MPO in immune responses remain debatable. In humans, genetic MPO deficiency is found in one individual out of two to four thousand, with varying subtypes of deficiency ranging from the total lack of expression to mildly expressed MPO levels [27]. Nevertheless, impaired immune responses are not observed in these individuals.

The first description of human MPO deficiency dates back to 1969, when a single otherwise healthy patient was described with a disseminated *C. albicans* infection that could be linked to the lack of myeloperoxidase activity in myeloid cells [28]. A subsequent study compared the response of MPO-deficient neutrophils and neutrophils from patients with the chronic granulomatous disease (CGD) lacking NADPH oxidase to *C. albicans*, showing that the postphagocytic oxidative burst was completely disrupted CGD [29]. MPO deficient leukocytes still mounted an efficient oxidative response and a candida-static effect [30]. Other cases of prolonged infection have been reported since then [31]. The first larger epidemiological study compared 92 cases of MPO deficient individuals to a matched control group and revealed an association of MPO deficiency with increased occurrence of infectious complications and higher prevalence of chronic inflammatory disease, specifically arthritis [32]. However, other studies [13,33] uncovered that the majority of MPO deficient individuals did not display clinical signs of immune suppression. This suggests a minor role of MPO for microbicidal activity and indicates compensatory mechanisms to overcome the deficiency. Tseng et al. provided evidence for a compensatory mechanism by increased extracellular H_2_O_2_ production and surface-level αϺβ2 integrin expression of murine MPO deficient neutrophils [34]. Thus, it has been described that murine MPO deficient neutrophils exhibited a significant increase in extracellular H_2_O_2_ and surface level of αϺβ2 integrin [34]. Moreover, Stendahl and colleagues [35] revealed enhanced phagocytic activity in MPO-deficient neutrophils, which was downregulated when the active enzyme was added. The authors suggest in addition to microbicidal function, the MPO–H_2_O_2_–halide system modulates the inflammatory response by impairing certain receptor-mediated recognition mechanisms of phagocytic cells, which otherwise could elicit inflammatory reactions and tissue injury. Schürmann et al. [36] demonstrated in a comprehensive murine study colocalization of MPO with the pathogen’s membrane in the phagosome. This enhanced the anti-bactericidal effects of MPO by focusing oxidative stress and local HOCl production, which in comparison to H_2_O_2_ has a considerably smaller radius of action. The authors, in contradiction to results from studies in cardiovascular research, showed that MPO does not amplify the oxidative immune response per se, but confines it to specific locations. Yet, circulating, transcytosed or endothelial bound MPO still imposes damage upon vessels and parenchymal tissue; however, because of the short life of oxidative MPO products, molecular interaction partners must be found in close proximity.

Surprisingly, the lack of MPO or its inhibition elevates T cell responses in lymph nodes, causing enhanced skin delayed-type hypersensitivity and antigen-induced arthri-tis [37]. Here, the inhibition of dendritic cell activation presumably occurs via MPO-generated reactive intermediates which subsequently leads to reduced T cell responses, if any.

However, high levels of MPO correlate with the prevalence of certain autoimmune diseases. Downregulation of MPO expression has been associated with incidence and se-verity of lupus erythematodes [38], Kawasaki’s disease [39], inflammatory bowel disease [40] and eosinophilic granulomatosis [41]. About half of the patients with eosinophilic granulomatosis showed perinuclear anti-neutrophil cytoplasmatic antibodies (pANCA) directed against MPO. The role of pANCAs in pathogenesis is a matter of debate, as is the role of MPO itself in this context. Although of primarily diagnostic value, animal models suggest a substantial contribution of pANCA to the progression of eosinophilic granulomatosis [42,43].

## 2. Myeloperoxidase (MPO) Regulation of Gene Expression

MPO is encoded on chromosome 17 in the human genome [44] and on chromosome 11 in the mouse genome (Figure 2a), with two protein-coding transcript isoforms of MPO produced in humans and four in mice [2]. Importantly, MPO gene expression is restricted to myeloblastic and the promyelocytic stages of myeloid lineage differentiation [45,46,47]. Interestingly, the gene is not expressed at earlier stages [48,49]. Due to its stage and lineage-specific transcription, MPO levels are used as a marker of normal or abnormal myeloid differentiation [50]. The tightly regulated transcription of this gene occurs in humans from three distinct promoters, however, only promoter 1 is contributing to the generation of a full-size transcript in vivo and is regulated during myeloid differentiation [51]. Data by Chumakov et al. [50] showed that the MPO promoter in human myeloid cells is located in the 5′-flanking region of the MPO gene between two Alu regions at bp −1648 to −1345 and at bp −531 to −200. More recent data by Lin et al. [51] suggested that the transcripts are initiated at three distinct sites within and downstream from this region. The promoter activity of the MPO gene in normal myeloid differentiation is mainly positively regulated by the transcription factors RUNX1 and CEBPA (Figure 2a) [52]. Similarly, transcription factors from the CEBP family promote the expression of other granule-associated enzymes and their abundance is tightly regulated throughout neutrophil generation [53]. CEBPA induces the expression of primary granule enzymes such as MPO, whereas CEBPE and CEBPD promote the expression of secondary and tertiary granule enzymes, such as LTF and MMP8, respectively [54,55]. The expression of these transcription factors was highly coordinated and the expression pattern correlated with granule expression and neutrophil development [53]. Primary granules containing MPO were mainly found at the granulocyte-macrophage progenitor stage where CEBPA is highly expressed, secondary granules formed mostly within bone marrow-committed proliferative neutrophil precursors and non-proliferative immature neutrophils expressing CEBPE, and tertiary granules were associated with mature neutrophils, where CEBPD was abundant. However, in acute myeloid leukemia (AML) caused by clonal expansion of early myelocytes insensitive to normal cellular signals, MPO expression is imbalanced. Changes in the methylation of CpG island #13 of the MPO gene are responsible for the dysregulated expression. The methylation pattern of the MPO gene is under control of the de novo methyltransferase DNMT3B, whose expression is negatively correlated with the abundance of MPO mRNA in AML [56]. Additionally, mutations in the first half of the upstream Alu element sequence create strong binding sites for the transcription factor SP1 leading to an almost 25-fold increase in MPO expression in the majority of AML patients [57]. The ANCA-associated vasculitis (AAV) promoter region of the MPO gene is depleted of transcriptional histone repression marks such as H3K27me3 and H3K9me2 and enriched in H4K16ac, a marker of active transcription. Accordingly, the promoter region in AAV is usually hypermethylated at CpG #13 [58,59] suggesting an important role of epigenetic control of MPO expression in this disease and the importance of methyltransferases EHMT1 and EHMT2 as well as components of histone acetyltransferase complex ING4 and MSL1 [58].

Mouse MPO transcription is initiated from four distinct promoter sites. The murine MPO promoter region lacks Alu sequences located in the 5′-flanking region of the gene [60]. Yet, a 1.3-kb promoter region of murine MPO shares sequence homology with the human gene. The region −315 to −264 contains a binding site for a transcriptional activator MyNF1, expressed in murine myeloblasts, but not erythroid, B-, or fibroblast cell lines [61], which is most likely responsible for the lineage-specific transcription of MPO in mice.

## 3. Atypical Expression of MPO in Human Disease: Cancer and Neurodegenerative Disorders

In homeostatic conditions, MPO is almost exclusively expressed in neutrophils and monocytes (Figure 2b). Expression by non-myeloid cells causes severe tissue damage and a potentially carcinogenic microenvironment [62,63]. Particularly, a study of human ovarian cancer detected a polymorphism associated with elevated MPO expression in several low-stage carcinomas, suggesting a role of MPO-mediated oxidative burst in cancer development (Figure 2b) [64]. The same polymorphism is associated with MPO expression in microglia and CNS macrophages in multiple sclerosis [65]. The carcinogenic environment is enhanced by the enzymatic activity of MPO and the resulting oxidants promote several axes of carcinogenesis [66]. MPO-derived reactive superoxides, specifically HOCl, impose DNA damage by directly modifying nucleotides and the formation of chloramines when reacting with histones. Lipid peroxidation, a highly established feature of HOCl, also contributes to the accumulation of DNA damage by the formation of malondialdehyde (MDA). Additionally, MPO has been shown to facilitate biotransformation and activation of carcinogens, specifically polycyclic aromatic hydrocarbons present in tobacco smoke. In accordance, a polymorphism of the MPO gene downregulating its expression is associated with decreased lung, breast and ovarian cancer risk [67,68]. Moreover, MPO can facilitate tumor progression and metastasis through upregulating matrix metalloproteinase (MMP) activity [66].

The double-edged sword of MPO activity enhances innate immunity in the process of cancer cell elimination, but simultaneously imposes mutagenic potential and disrupts the extracellular space, allowing tumors to infiltrate their surroundings and metastasize more rapidly.

Particularly harmful are high levels of MPO in post-mitotic tissues with limited capacity to regenerate, such as the central nervous system (CNS) (Figure 1c) [69,70]. Although, the blood–brain barrier (BBB) of the CNS restricts transmigration of leukocytes under physiological conditions [71], during the acute inflammatory response and several chronic pathologies, the permeability of the BBB increases [72], partially due to the HOCl produced by MPO of circulatory polymorphonuclear leukocytes [73]. Within the CNS, MPO is expressed mostly in microglia, the residential macrophages of the brain parenchyma [65], although a recent study detected MPO expression in neurons [74]. Increased levels of neuronal MPO have specifically been linked to neurodegenerative diseases. Thus, in Parkinson’s Disease, where neuronal expression of MPO in the substantia nigra is increased [75], the formation of chlorodopamine by HOCl induces cell death of dopaminergic neurons, the central underlying process of this disease [76]. Of importance, mice deficient for MPO are more resistant to the parkinsonian neurotoxin 1-methyl-4-phenyl-1,2,3,6-tetrahydropyridine [77].

Likewise, neurons from patients with Alzheimer’s Disease (AD) expressed increased MPO levels [74] and MPO is co-localized with amyloid-b protein, which is crucially involved in AD [78]. MPO deficient animals are partially rescued from AD [79].

## 4. Enzymatically Driven Role in Cardiovascular Disorders

Currently, the main impact of myeloperoxidase on cardiovascular diseases is related to derivatives of its enzymatic activity (Figure 2b and Figure 3). Many interesting clinical observations of MPO and cardiovascular health have been reported. For example, elevated levels of leukocyte- and blood-MPO are associated with the presence of coronary artery disease (CAD) [80] and plasma MPO may predict the long-term risk of cardiovascular mortality in patients with CAD [81,82]. Multiple epidemiological studies found MPO deficient individuals to be protected from CHD and other clinical manifestations of atherosclerosis. Basic and clinical research pinpointed several mechanisms behind this association and could successfully establish the enzyme as an important player in different types of cardiovascular disease. Numerous studies, including works from our group, suggest that inhibition of MPO activity is beneficial for the treatment of cardiovascular diseases. The strong oxidizing capacity of MPO at physiological pH might facilitate its activity not only inside of the acidic phagosomes of neutrophils, but also in tissue. This contributes to the chlorination of proteins and other small molecules outside of the neutrophils and can impact cardiovascular diseases through the modification of inflammatory mediators, for example, MMPs, which play a pivotal role in vascular and cardiac remodeling through proteolysis of extracellular matrix [83]. Thus, MPO-derived HOCl may regulate proteolytic events at sites of acute inflammation by activating proteases such as MMP-7 [84,85]. The hypothesis of an oxidative enzyme “gone rogue” in conjunction with the aforementioned strong epidemiological evidence for an inversed correlation between MPO activity and the risk of CHD prompted the generation of an MPO-deficient mouse strain to study underlying mechanisms. The first description of MPO mutant mice by Brennan and colleagues surprisingly yielded anti-atherosclerotic effects by MPO [86]. Low density lipoprotein (LDL) receptor-deficient (Ldlr^−/−^) mice both cross-bred with Mpo^−/−^ mice or reconstituted with MPO-deficient bone marrow after irradiation harbored severely advanced atherosclerotic plaques compared to controls [86]. This finding contradicted previous knowledge and hypotheses about the enzyme and its role in cardiovascular diseases. The same study did not detect MPO expression in plaques of Ldlr^−/−^ mice, whereas MPO activity within the lesion is commonly found in human atherosclerotic plaques and thought to be a major facilitator of disease progression. This difference was explained partly by pointing out differences of both MPO and pathogenesis of atherosclerosis between humans and mice. Since in mice the neutrophil/lymphocyte ratio is inversed and MPO levels per neutrophil are five- to tenfold lower compared to humans, the biological impact of MPO might be generally underestimated when using mouse models. Additionally, the role of MPO in murine and human biology may differ as neutrophils of MPO deficient mice still possess residual candidacidal properties. The difference in plaque composition might also be based on the nature of the disease progression in mice and humans: whilst atherosclerosis develops for decades in humans, already young, genetically modified mice have atherosclerotic lesions. For some time, cardiovascular research on MPO in rodents focused on models of inflammatory remodeling in which neutrophils and MPO was present and active [87]. Nearly 20 years later, another study partially resolved the conundrum of MPO in murine atherosclerosis. MPO deficiency and enzymatic inhibition in Apoe^−/−^ mice increased atherosclerotic lesion size similar to the results obtained in Ldlr^−/−^ mice, but the atherosclerotic plaques shifted to a stable phenotype with a thick fibrous cap when MPO was absent or inhibited [88]. Although an increased lesion size per se can become symptomatic in drastic cases, plaque instability is a much stronger predictor of adverse events—that is, plaque rupture and succeeding thromboembolism—in atherosclerosis patients [89]. Additionally, the study conducted in Apoe^−/−^ mice could detect MPO activity within plaques in vivo by visualizing its oxidative products with MRI [88]. The lack of MPO activity in plaques of Ldlr^−/−^ mice might either be due to specific shortcomings of that model or the lack of sufficiently sensitive methods at that time.

The mouse model and, eventually, the development of experimental therapeutics for MPO inhibition yielded additional insights into the role of the enzyme in cardiovascular pathologies that were accompanied by a growing body of clinical evidence. Besides atherosclerotic plaque formation, the enzyme possesses vasomotor properties, elevating systemic vascular resistance and interrupting endothelial function by direct and indirect nitric oxide consumption [15], accumulation of asymmetric dimethylarginine [90] and, more recently described, inhibition of soluble guanylatecyclase [91]. This effect was confirmed in a large animal and human study where MPO-deficient individuals were less likely to suffer from a nicotine-induced reduction of flow-mediated dilation and MPO infusion increased vascular tone [92]. It has also been shown that MPO binds to the vessel wall and causes oxidation of endothelium-derived nitric oxide (NO), leading to a decrease in NO bioavailability and resulting in endothelium-dependent vasorelaxation [93]. Thereby, it may contribute to the endothelial dysfunction that precedes manifest atherosclerotic lesions.

Additionally, MPO takes part in the formation of oxidized LDL (oxLDL), which after transcytosis through the defective endothelial barrier is phagocytosed by aortic macro-phages. Once their capacity to clear lipids is exceeded, these cells store lipids in vesicles, which coins them according to their microscopic appearance as foam cells. Whereas native LDL is only sporadically taken up by macrophages, oxLDL extensively provokes phagocytosis [94]. This integral step in the pathogenesis of atherosclerosis is promoted by the oxidative capabilities of MPO [95]. Both lipid peroxidation and posttranslational modification of protein components contribute to oxLDL formation [96,97]. MPO has been shown to catalyze both these processes. Lipid peroxidation is triggered by MPO in inflammatory tissue as a means to produce reactive oxidants [24]. Moreover, MPO modifies amino acid residues of Apolipoprotein B-100 (apoB-100), the major protein component of LDL. The resulting, highly pro-inflammatory complex has been branded Myeloperoxidase-oxidized LDL (MoxLDL). MPO localizes to the surface of LDL due to its cationic charge, resulting in the formation of superoxides in close proximity [98] and MPO-specific protein modifications can be found in human atheroma [99,100,101]. In a similar manner, MPO might contribute to oxidation of high-density lipoprotein (HDL) and specifically apolipoprotein A-I (apoA-I), limiting its atheroprotective properties [102]. HDL isolated from human atherosclerotic plaque displays MPO-specific posttranslational modifications [103,104]. However, this effect is less established than the interactions with LDL and a study conducted in a Chinese population could not associate MPO mediated oxidation of HDL and HDL dysfunction [105]. Apart from disruption of endothelial function and lipoprotein oxidation, MPO presence in atheroma can destabilize plaques by extracellular matrix degradation and platelet activation. A plethora of ECM proteins are susceptible to MPO-mediated modifications [106,107,108,109]. The enzyme also renders platelets thrombogenic by interactions with their surface and cytoskeleton that are not entirely understood [110]. The accumulation of coagulated material in atheroma is a widely recognized hallmark of plaque destabilization with platelet inhibition being the most potent pharmaceutical intervention against the progression of CHD.

Apart from its implication in multiple mechanisms of atherosclerotic pathophysiology, there is additional experimental evidence for a direct unfavorable influence of MPO on cardiac tissue homeostasis in the context of ischemic and hypertensive heart disease. The myocardial extracellular matrix is produced majorly by fibroblasts and to a lesser extent by cardiomyocytes and vascular smooth muscle cells. ECM degradation by MMPs triggers a release of proinflammatory cytokines that provoke transdifferentiation of fibroblasts to myofibroblasts [111]. This phenotype produces collagen and fibrotic scarring ensues. Although initially necessary for wound healing in the context of myocardial infarction (MI), fibrotic remodeling may turn maladaptive when it affects a substantial portion of the myocardium, increases ventricular stiffness and promotes the development of arrhythmia. In myocardial infarction, the local presence of MPO as part of the early neutrophil infiltration into the infarction zone argues for a role of the enzyme in post-infarction remodeling. This was confirmed in Mpo^−/−^ mice, which displayed retardation of fibrotic scar formation and an overall beneficial functional outcome after permanent LAD ligation induced MI [112]; of note, MPO does not affect total infarction size [112,113,114]. More recently, Ali and colleagues showed that the prolonged MPO inhibition with PF-1355 in mouse models of MI and ischemia-reperfusion injury (IRI) improved ejection fraction, end-diastolic/systolic volume and left ventricular (LV) hypertrophy. The authors propose that inhibition of MPO at later stages of heart remodeling may favor the healing process and improve chronic outcomes [115]. In a similar manner, tissue destruction in the wake of infarction promoted ventricular arrhythmia which did occur to a significantly lower extent in Mpo^−/−^ mice [114]. The latter finding complemented and extended a study by Rudolph and colleagues that presented MPO as an activator of MMPs in the context of overload-driven atrial fibrillation [85], further suggesting that adverse tissue remodeling by MPO in the context of cardiac disease has both hemodynamic and arrhythmogenic consequences.

Interestingly, patients with heart failure (HF) also have increased plasma MPO levels [116,117]. Deuschl and colleagues have shown that MPO deficient mice subjected to mild transverse aortic constriction as a model of LV hypertrophy show a significantly decreased hypertrophic reaction. The authors, therefore, claim that MPO is critically linked to impaired diastolic compliance during pressure overload and suggest that MPO is a potential pharmacological target in HF [118]. The positive effect on chronic LV remodeling is thought to be mediated through inhibition of MPO-oxidized aldehydes and through a decreased recruitment of pro-inflammatory cells [113] and through a decreased recruitment of pro-inflammatory cells [115]. Endothelial dysfunction as a mechanism of chronic heart failure progression could potentially lead to a refocusing on MPO as a therapeutical target for heart failure. Therefore, MPO deficiency or inhibition of its enzymatic activity is beneficial for cardiovascular morbidity [32,119].

There have been multiple strategies of MPO inhibition proposed, including small-molecule drugs, chelators, antioxidants, scavenger and natural products like flavonoids and polyphenols (reviewed here [120]). However, their selectivity and specificity and thus the potential of clinical applications remains debatable. The most selective MPO inhibitors are so-called “suicide” substrates, working in a mechanism-dependent manner, and causing irreversible heme destruction via generation of ferrous MPO, or modification of the heme groups. Implementation MPO inhibitors, such as 4-aminobenzoic acid hydrazide (ABAH) has been shown to reduce infarct size and neuronal deficit in a murine model of stroke [121], whereas AZM198, a thioxanthine derivative produced by AstraZeneca was used to inhibit MPO activity in atherosclerotic plaque model [88] and pulmonary arterial hypertension [122].

## 5. Myeloperoxidase Is Important for NETs Formation

Apart from the enzymatic role in reactive oxygen species (ROS) production and phagocytosis, MPO is actively participating in degranulation, and the generation of neutrophil extracellular traps (NETs), a process known as NETosis, where the positive charge of the protein is of high significance (Figure 4a).

NETs are a meshwork of chromatin fibers with a diameter of 15–17 nm. DNA and histones form the major content of NETs, however, a variety of other neutrophil-specific proteins of mainly proteolytic or microbicidal activity such as neutrophil elastase, myeloperoxidase, cathepsin G, azurocidin, lactotranferin, lysozyme C and others encrust NETs as well [123].

NET formation requires disintegration of nuclear and granule membranes from nuclei released of anucleated neutrophils or from neutrophils undergoing cell death [124,125,126]. NETosis can be triggered by treatment with phorbol myristate acetate (PMA), a specific activator of Protein Kinase C (PKC), and hence of nuclear factor-kappa B (NF-κB). Additional triggers for NETosis include interleukin 8 (IL-8), lipopolysaccharide (LPS), activated endothelial cells, nitric oxide and various autoantibodies. More stimuli are reviewed here [127,128].

MPO as part of NETs is an important component of the immune response to infection in vivo [129]. MPO on NETs remains enzymatically active at this site [130]. Growing evidence shows that MPO is necessary for PMA-stimulated NETosis [35]. PMA-driven NETs formation capacity correlates with the MPO content in neutrophil granules, and in mice maximum levels of NETosis were observed during night time [9]. However, the aiding role of MPO in NET formation depends on the trigger of NETosis. Thus, inhibition of MPO had no effect on NET formation when triggered by *Pseudomonas aeruginosa*, *Staphylococcus aureus*, or *Escherichia coli* [131]. The NET formation is induced by ROS production, generated by NADPH-oxidase (NOX) or mitochondria. Upon ROS production, neutrophil elastase (NE) and MPO stored in the rested azurophilic granules escape and translocate into the nucleus, where the elastase cleaves histones, particularly histone H4 and promotes chromatin decondensation [132,133]. Positively charged MPO enters the nucleus a little later when the chromatin is partially decondensed, presumably being attracted by the negatively charged free DNA. In the nucleus, MPO further facilitates chromatin decondensation. In addition to the NE activity, chromatin decondensation is facilitated by the activity of peptidylarginine deiminase PAD4, which converts positively-charged arginine residues of histones into citrullins [134] and reduces the positive charge of proteins [135]. Moreover, histones get acetylated during NET formation [136]. This modification as well as citrullination further promotes chromatin decondensation and transcription firing, supporting rapid and irreversible NETosis [136]. Of note, granule content, changes in proteome and NET-forming activity in human neutrophils were associated with variations in the susceptibility to develop lung inflammation and its magnitude [9].

## 6. Non-Enzymatic Role of MPO

The positive charge of MPO was shown to recruit PMNs and evoke PMN motility solely dependent on the electrostatic interactions with the leukocyte’s surface [129]. Another non-enzymatic role of MPO has been recently described by Manchanda with colleagues [137] (Figure 4b). The authors show that MPO treatment causes a collapse of the endothelial glycocalyx via binding with heparan sulfate and causes shedding of syndecan1 in a neutrophil-dependent manner (Figure 4b). This property of MPO is independent of its catalytic function and is purely caused by the nature of its cationic charge. Due to the highly cationic surface charge, MPO binds to the anionic surface of red blood cells. The catalytic activity of MPO persists, thereby red blood cells can serve as carriers of competent MPO to remote organs or vasculature. The charge-driven affinity of MPO can be reversed by heparin administration [138].

Along with the capacity to bind negatively charged cellular membranes, MPO is able to transmigrate into various cell types, such as human umbilical vein endothelial cells (HUVECs) as well as into other endothelial (ECs) and epithelial cells [137,139]. In less than 6 h, MPO can be detected on the cell surface, in the cytoplasm and within the nucleus of ECs. MPO is also readily internalized through the mannose 6-phosphate receptor and is delivered to lysosomes of retinal pigmented epithelial (RPE) cells (Figure 4c) [140]. When deposited in lysosomes of RPEs, MPO can both stimulate autophagy, triggering nuclear translocation of autophagy specific transcription factor TFEB and cause the clearance of retinal lipofuscin deposits and, when chronic treatment with MPO performed, cause lysosomal stress and cell death [140].

Interestingly, in the microenvironment of an inflammatory response, approximately 40% of MPO is rapidly inactivated by oxidation [141]. Edwards et al. demonstrated the presence of 16–29 mg/mL of enzymatically inactive MPO (iMPO) in an arthritic joint [33]. Therefore, at the site of inflammation, both active and inactive MPO are present. Unexpectedly, the inactive form of MPO as well as its enzymatically active counterpart [142] cause transcriptional changes in ECs [143]. Among others, the expression and secretion of cytokines including interleukin (IL)-6 and -8, MCP-1, and GM-CSF increased rapidly and in a dose-time-dependent manner [143]. The selective overexpression of the pro-inflammatory factors in response to MPO treatment may suggest that MPO is a novel potential trigger of endothelial cell activation.

The above-mentioned examples underline the importance of the yet poorly studied non-catalytic activities of MPO. At least three potential different mechanisms of non-enzymatic role of MPO might be of interest: (1) cationic MPO changes a negatively charged cellular surface and unmasks receptors on it, (2) when entering the cell, MPO might affect the autophagy flux and cause enhanced transcription of inflammatory genes through the activation of transcription factor TFEB, (3) when entering the nucleus, MPO might affect transcription directly by binding to the DNA, histones, or particular transcription factors.

## 7. Conclusions

The signature properties of MPO and its function in different cell types and tissues is connected to its enzymatic activity and the generation of reactive oxygen species. As a partaker in the immune response, MPO demonstrates beneficial effects in anti-fungal responses or phagocytic activity of neutrophils and the hyperactivation of dendritic cells, however, increased MPO levels are also associated with autoimmune diseases such as lupus erythematodes and Kawasaki’s disease. The produced oxidants possess microbicidal properties, but are also engaged in oxidative tissue damage.

Therefore, MPO expression is tightly regulated in a lineage-specific manner and dysregulation of its expression has been linked to multiple diseases, including cancer and neurodegenerative disorders. MPO expression and its enzymatic activity in the blood vessel seem to be detrimental for the progression of cardiovascular diseases.

Recent studies accumulate knowledge on the non-enzymatic role of MPO in vessels and beyond. Due to its cationic charge, it can severely affect the endothelial glycocalyx properties and change the expression profile of endothelial cells, presumably causing endothelial dysfunction. Various cell types can internalize MPO, which leads to diverse and poorly studied mechanisms of cellular transformations, for example through autophagy activation. After prolonged administration, MPO can enter the nucleus of endothelial cells, where its role has not been studied so far.

Through enzymatic and non-enzymatic capacities, MPO becomes an important contributor to the local regulation of immune responses in a variety of tissues. The over-whelming data on adverse effects of MPO in cardiovascular disease with respect to only few described beneficial functions raise the question of whether the physiological properties of the enzyme have been adequately explored and described. A better understanding of distinguishing useful and potentially necessary functions from excessive tissue damage may be useful in developing new therapeutic strategies for inflammatory diseases. Broad modulation of host-immunity is regularly accompanied by significant side effects, thus a leukocytic enzyme that shows predominantly negative effects and appears to have no ob-ligate functions is an attractive pharmacological target.

Despite longstanding research on MPO, there is little understanding about the mo-lecular outcomes of MPO activity or its inhibition in vivo, in different tissues and during disease progression. Thus, additional work with the integration of several omic-approaches, such as single-cell transcriptomics, proteomics, epigenomics performed on mice and human tissues, as recently applied by Li et al. [144], is needed for discriminating the desired and beneficial effects of the protein from its detrimental activities. Targeted activation or inhibition of downstream affected pathways, rather than general blocking of the enzymatic activity may help to preserve the positive effects without sacrificing a potential effector of host defense. Additionally, further work should be performed using bioinformatic approaches to integrate complex parameters of disease progression with multiple related genetic, epigenetic and environmental factors to better determine the nature of the association of MPO deficiency and improved cardiovascular health.

## Figures and Tables

**Figure 1 antioxidants-10-00562-f001:**
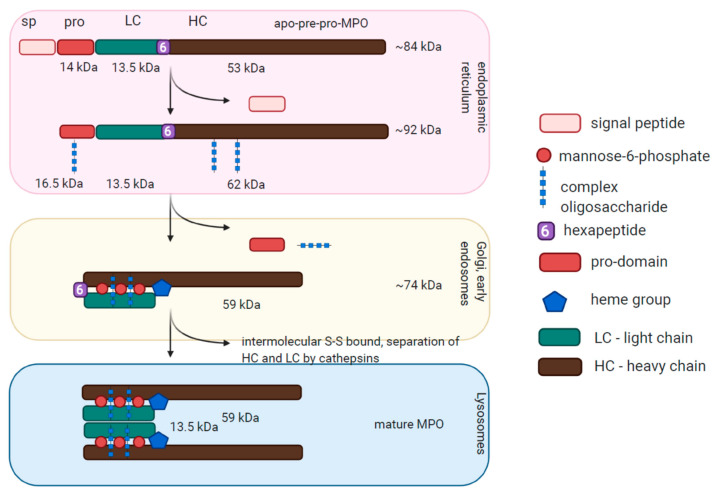
Structure and maturation of Myeloperoxidase. Myeloperoxidase (MPO) is translated as an 84 kDa apo-pre-pro-MPO protein and undergoes several maturation steps. The first step of maturation takes place in the endoplasmic reticulum, where proteins acquire post-translational modifications and lose the signal peptide. The second step occurs in the Golgi apparatus and early endosomes, where the heme group is acquired and the pro-domain is lost. The third step takes place in lysosomes, where the intermolecular S–S bounds are formed and heavy and light chains of MPO are separated with the formation of mature MPO. Adapted from Laura et al. [10].

**Figure 2 antioxidants-10-00562-f002:**
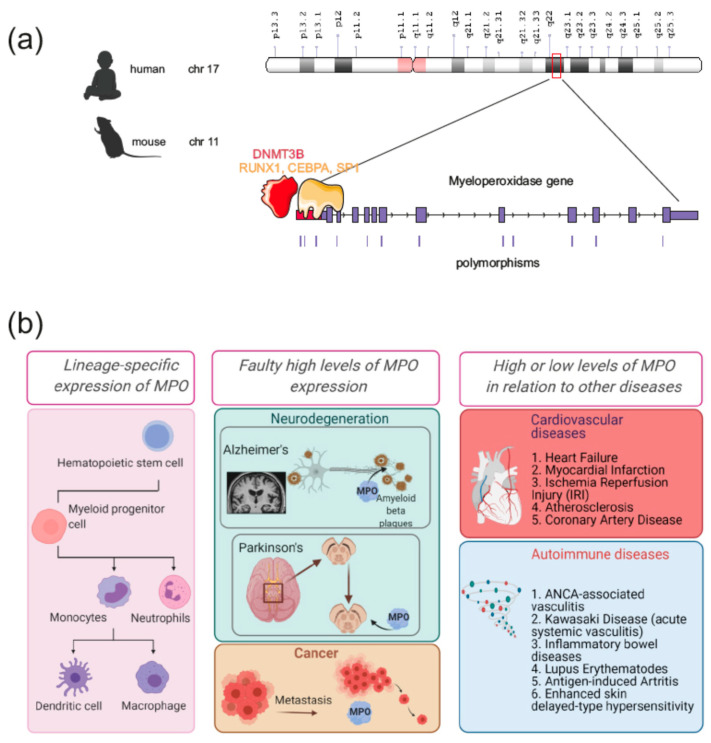
Myeloperoxidase gene expression regulation in health and disease. (**a**) Myeloperoxidase is located on chromosome 17 of the human genome and chromosome 11 of the mouse genome. MPO expression is tightly regulated in a lineage-specific manner by transcription factors RUNX1, CEBPA, SP1 and on the epigenetic level by variations in DNA and histone methylation patterns. (**b**) MPO expression is tightly regulated in a lineage-specific manner. De-regulation of MPO expression is associated with cancer and neurodegenerative diseases, whereas cardiovascular diseases are usually associated with increased MPO levels, and autoimmune diseases are associated with both too high and too low levels of MPO.

**Figure 3 antioxidants-10-00562-f003:**
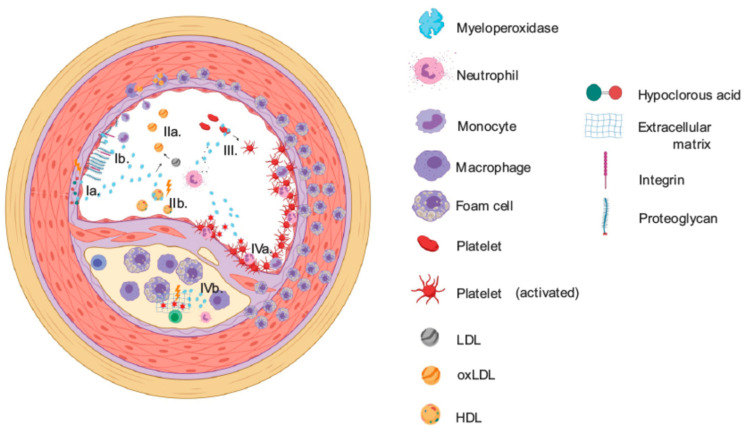
MPO as a mediator of vascular disease. MPO affects endothelial integrity and function by reducing NO bioavailability in the subendothelial space (Ia.) and collapsing the endothelial glycocalyx, compromising mechanosensitivity and exposing leukocyte receptors (Ib). The enzyme further oxidizes low-density lipoprotein (LDL) (IIa.) and modifies high-density lipoprotein (HDL) (IIb.) with respective atherogenic consequences. MPO-mediated platelet activation contributes to the formation of atheroma (III.). Within the plaque, active MPO is found to destabilize cell adhesion and extracellular matrix (IV.), which promotes plaque rupture.

**Figure 4 antioxidants-10-00562-f004:**
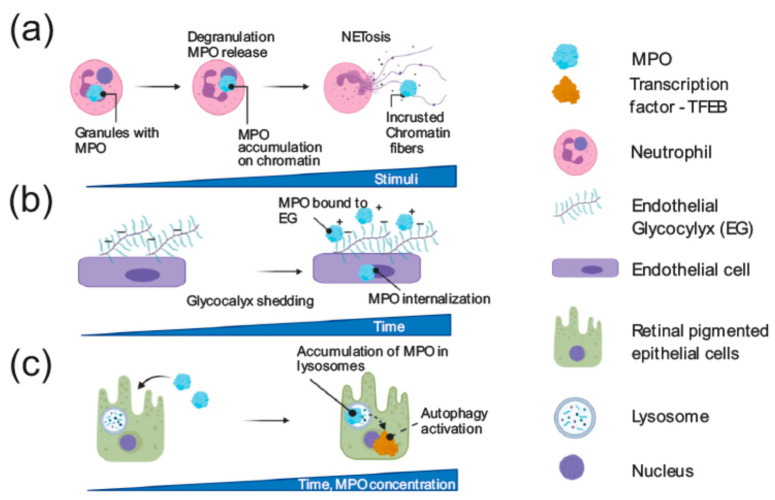
Neutrophil extracellular trap (NET) formation and non-enzymatic modes of MPO effects in different cell types. (**a**) MPO is important for NET formation. Stored in azurophilic granules of rested neutrophils, MPO is released upon various stimuli, contributes to the formation of NETs, and becomes part of NETs when encrusting the chromatin. (**b**) Due to its cationic charge, MPO shows a high affinity towards negatively charged endothelial glycocalyx. Moreover, MPO is internalized by endothelial cells. (**c**) In retinal pigmented epithelial cells MPO is internalized and accumulated in lysosomes leading to activation of autophagy.

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
