# Peer review of "The Enzymatic and Non-Enzymatic Function of Myeloperoxidase (MPO) in Inflammatory Communication"

_antioxidants, 2021, doi:10.3390/antiox10040562_

Round 1
Reviewer 1 Report
In this paper Kargapolova and colleagues reported the MPO known role in inflammation and how it can influence disease progression from cancer to cardiovascular diseases.
I found this paper interesting and I believe it can provide good new insights in the field.
Therefore, I suggest its publication.
I only recommend minor changes to improve the manuscript:
1) The “inter-organ communication” mentioned in the title is not clear to me.
Authors describe the functionality of MPO in the immune cells and how that influences the immune response and the progression of several diseases. However, the mentioned inter-organ communication is not described enough along the work. If they want to keep this title, they should improve that part defining how MPO function in different organs can be interconnected.
2) The authors nicely described the function of MPO in cardiovascular disease, mostly suggesting that inhibition of MPO activity is beneficial for treatment of cardiovascular diseases.
However, initial studies with MPO KO mice suggested an opposite function of MPO having more atherosclerosis progression compared to WT (Brennan et al. J Clin Invest. 2001 Feb 15; 107(4): 419–430.). This was attributed to few levels of MPO in mice plaques, also suggesting that human and mice MPO might have different functions in pathology (Hazen Arteriosclerosis, Thrombosis, and Vascular Biology. 2005;25:1102–1111). Moreover, the actual functionality of current MPO inhibitors in humans is still under debate.
I believe authors should also mention and discuss contrasting data in the field.
3) Another important section is the Lipid Peroxidation induced by MPO. This is quite interesting and would be nice to improve its discussion. Especially considering its role in cardiovascular diseases and atherosclerosis.
4) it would be also helpful to include a general figure highlighting the known involvement of MPOs in all the disease discussed possibly expanding Figure 2b.
Author Response
Reviewer #1
Remarks to the Author:
In this paper Kargapolova and colleagues reported the MPO known role in inflammation and how it can influence disease progression from cancer to cardiovascular diseases.
1) The “inter-organ communication” mentioned in the title is not clear to me.
Authors describe the functionality of MPO in the immune cells and how that influences the immune response and the progression of several diseases. However, the mentioned inter-organ communication is not described enough along the work. If they want to keep this title, they should improve that part defining how MPO function in different organs can be interconnected.
Author response:
We agree with the reviewer in regards to rather brief touch on the topic of “inter-organ communication”, and changed the title of the paper to be:
“The enzymatic and non-enzymatic function of myeloperoxidase (MPO) in inflammatory communication”
We hope the new title reflects entirely the content of our publication.
2) The authors nicely described the function of MPO in cardiovascular disease, mostly suggesting that inhibition of MPO activity is beneficial for treatment of cardiovascular diseases.
However, initial studies with MPO KO mice suggested an opposite function of MPO having more atherosclerosis progression compared to WT (Brennan et al. J Clin Invest. 2001 Feb 15; 107(4): 419–430.). This was attributed to few levels of MPO in mice plaques, also suggesting that human and mice MPO might have different functions in pathology (Hazen Arteriosclerosis, Thrombosis, and Vascular Biology. 2005;25:1102–1111). Moreover, the actual functionality of current MPO inhibitors in humans is still under debate.
I believe authors should also mention and discuss contrasting data in the field.
Author response:
We appreciate the very positive feedback of the reviewer and are thankful for their suggestions on the cardiovascular part of the review. As suggested, we now included the contrasting data in human and mouse experiments by Brennan et. al. 2001, as well as Nicholls and Hazen 2005, references 84 and 85 accordingly. We are hopeful that we now addressed the discrepancies between the role of MPO in plaque formation in mice and human to the fullest, additional text can be found on page 7.
Indeed although MPO inhibition was mentioned by us before, the debatable nature of the activity of inhibitors and their applicability in clinics was not clearly stated. Therefore, we also added additional information on MPO inhibitors, which can be found on pages 9-10.
3) Another important section is the Lipid Peroxidation induced by MPO. This is quite interesting and would be nice to improve its discussion. Especially considering its role in cardiovascular diseases and atherosclerosis.
We thank the reviewer for pointing to this important section, which was minimized in the initial version on the manuscript. We extended our discussion on Lipid Peroxidation induced by MPO, which can be found on pages 8-9.
4) it would be also helpful to include a general figure highlighting the known involvement of MPOs in all the disease discussed possibly expanding Figure 2b.
We agree with the reviewer, a general overview of the known involvement of MPO in the diseases would help the reader to estimate the spectrum of diseases mentioned in the review. We therefore changed Figure 2b accordingly.
We are very thankful for the thoughtful comments and recommendations of the reviewer and hope that we have carefully addressed their suggestions and in doing so feel the manuscript is substantially strengthened.
Reviewer 2 Report
In the review “The enzymatic and non-enzymatic function of myeloperoxidase (MPO) in
inflammation and inter-organ communication” the authors review the current literature on myeloperoxidase and its role played enzymatically and non-enzymatically in the immune response. Also, the implication of the myeloperoxidase in cancer, neurodegenerative disease and cardiovascular disorders are discussed.
The review is well written and ready for publication. Summary figures strengthening the message are given.
Author Response
Remarks to the Author:
In the review “The enzymatic and non-enzymatic function of myeloperoxidase (MPO) in inflammation and inter-organ communication” the authors review the current literature on myeloperoxidase and its role played enzymatically and non-enzymatically in the immune response. Also, the implication of the myeloperoxidase in cancer, neurodegenerative disease and cardiovascular disorders are discussed.
The review is well written and ready for publication. Summary figures strengthening the message are given.
Author response:
We are glad to see that the reviewer found our manuscript well written and ready for publication.